# Model-Agnostic Meta-Learning for Multimodal Task Distributions

## Abstract

Gradient-based meta-learners such as MAML (Finn et al., 2017) are able to learn a meta-prior from similar tasks to adapt to novel tasks from the same distribution with few gradient updates. One important limitation of such frameworks is that they seek a common initialization shared across the entire task distribution, substantially limiting the diversity of the task distributions that they are able to learn from. In this paper, we augment MAML with the capability to identify tasks sampled from a multimodal task distribution and adapt quickly through gradient updates. Specifically, we propose a multimodal MAML algorithm that is able to modulate its meta-learned prior according to the identified task, allowing faster adaptation. We evaluate the proposed model on a diverse set of problems including regression, few-shot image classification, and reinforcement learning. The results demonstrate the effectiveness of our model in modulating the meta-learned prior in response to the characteristics of tasks sampled from a multimodal distribution.

## 1 Introduction

Humans are capable of effectively utilizing prior knowledge to acquire new skills as well as adapt to new environments. For example, imagine that we are learning how to carve on a snowboard on a mountain that we just arrived at. While we only know the basics of snowboarding and barely know the terrain, we can still accomplish this feat quickly. Even when the skill that we are interested in learning is only vaguely related to the set of skills that we have acquired in the past, we are usually able to exploit the relationship among skills and generalize. For the snowboarding example, our knowledge about skiing and skateboarding can play the role that prepares us for learning snowboarding skills more efficiently.

Similarly, can machines rapidly master a novel skill based on a variety of related skills they have already acquired? Recent advances in meta-learning offer machines a way to rapidly adapt to a new task using few samples by first learning an internal representation that matches similar tasks. Such representations can be learned by considering a distribution over similar tasks as the training data distribution. Model-based meta-learning approaches (Duan et al., 2016; Wang et al., 2016; Munkhdalai & Yu, 2017; Mishra et al., 2018) propose to recognize the task identity from a few sample data, use the task identity to adjust a model's state (*e.g.* RNN's internal states) and make the appropriate predictions with the adjusted model. Those methods demonstrate good performance at the expense of having to hand-design architectures, yet the optimal strategy of designing a meta-learner for arbitrary tasks may not be obvious to humans. On the other hand, model-agnostic gradient-based meta-learning frameworks (Finn et al., 2017; 2018; Kim et al., 2018; Lee & Choi, 2018; Grant et al., 2018) seek an initialization of model parameters such that a small number of gradient updates will lead to high performance on a new task, while having flexibility in the choice of models.

While most of the existing gradient-based meta-learners rely on a single initialization, different tasks from the task distribution can require substantially different parameters, making finding an initialization that is a short distance away from all of them infeasible. If the task distribution is multimodal with disjoint and far apart modes, one can imagine that a set of separate meta-learners with each covering one mode could better master the full distribution. However, associating each task with one of the meta-learners not only requires additional identity information about the task, which is not always available or could be ambiguous when the modes are not clearly disjoint, but also eliminates the possibility of transferring knowledge across different modes of the task distribution.

To overcome this issue, we aim to develop a meta-learner that is able to acquire a meta-learned prior and adapt quickly given tasks sampled from a multimodal task distribution.

To this end, we leverage the strengths of the two main lines of existing meta-learning techniques: model-based and gradient-based meta-learning. Specifically, we propose to augment gradient-based meta-learners with the capability of generalizing across a multimodal task distribution. Instead of learning a single initialization point in the parameter space, we propose to first compute the task identity of a sampled task by examining task related samples. Given the estimated task identity, our model then performs a step of *model-based adaptation* to condition the meta-learned initialization on the sampled task in a step we call modulation. Then, from this modulated meta-prior, a few steps of *gradient-based adaptation* are performed towards the target task to progressively improve the performance on the task.

The main contributions of this paper are as follows:

- We identify and empirically demonstrate a limitation in a family of widely used gradient-based meta-learners, including Finn et al. (2017; 2018); Kim et al. (2018); Lee & Choi (2018); Grant et al. (2018); Nichol & Schulman (2018).

- We propose a model that is able to acquire a set of meta-learned prior parameters and adapt quickly given tasks sampled from a multimodal task distribution by taking advantage of both model-based and gradient-based meta-learning.

- We design a set of multimodal meta-learning problems and demonstrate that our model offers a better generalization ability on tasks including regression, classification, and reinforcement learning.

## 2 RELATED WORK

We review the several families of meta-learning, which our work builds upon:

**Optimization-based meta-learning.** The approaches that learn to optimize a learner model are known as optimization-based meta-learning. Pioneered by Schmidhuber; Bengio et al. (1992), optimization algorithms with learned parameters have been proposed, enabling the automatic exploitation of the structure of learning problems. From a reinforcement learning perspective, Li & Malik (2016) represent an optimization algorithm as a learning policy. Andrychowicz et al. (2016) train LSTM optimizers to learn update rules from the history of gradients, and Ravi & Larochelle (2016) train a meta-learner LSTM to update a learner's parameters for few-shot image classification. Unlike these methods, our algorithm does not require additional parameters to learn update rules; instead, we augment our meta-learner with the ability to compute the identity of tasks.

**Model-based meta-learning.** Model-based meta-learning frameworks learn to recognize the identities of tasks and adjust the model state (*e.g.* the internal state of an RNN) to fit the task. Santoro et al. (2016) train a network with an external memory that is able to assimilate new samples and leverage this data to make accurate predictions. Duan et al. (2016) represent a fast RL algorithm as an RNN and learn it from data. Munkhdalai & Yu (2017) propose to learn task-agnostic knowledge and use it to shift its fast parameters for rapid generalization. Though our model has a model-based meta-learning component, the main adaptation mechanism is gradient-based, which has preferable behavior on tasks outside the task distribution (Finn & Levine, 2018).

**Gradient-based meta-learning.** Finn et al. (2017) and its extensions (Finn et al., 2018; Kim et al., 2018; Lee & Choi, 2018; Grant et al., 2018), known as gradient-based meta-learning, aim to estimate a parameter initialization among the task-specific models, that provides a favorable inductive bias for fast adaptation. With the model agnostic nature, appealing results have been shown on a variety of learning problems. However, assuming tasks are sampled from a concentrated distribution and pursuing a common initialization to all tasks can substantially limit the performance of such methods on multimodal task distributions where the center in the task space becomes ambiguous. In this paper, we propose to first identify the mode of a sampled task, in a procedure which is similar to model-based meta-learning approaches Santoro et al. (2016); Mishra et al. (2018). Then, we modulate the meta-learned prior in the parameter space to make the model better fit to the mode and take gradient steps to rapidly improve the performance on the task afterwards.

## 3 PRELIMINARIES

We aim to rapidly adapt to a novel task sampled from a multimodal task distribution. We consider a target dataset $\mathcal{D}$ consisting of tasks sampled from a multimodal distribution. The dataset is split into meta-training and meta-testing sets, which are further divided into task-specific training $\mathcal{D}_{\mathcal{T}}^{\text{train}}$ and validation $\mathcal{D}_{\mathcal{T}}^{\text{val}}$ sets. A meta-learner learns about the underlying structure of the task distribution through training on the meta-training set and is evaluated on meta-testing set.

Our work builds upon Model-Agnostic Meta-Learning (MAML) algorithm (Finn et al., 2017). MAML seeks an initialization of parameters $\theta$ of a meta-learner such that after a small number of gradient steps to minimize the task-specific loss on the training set of a new task $\mathcal{D}_{\mathcal{T}}^{\text{train}}$, the adapted parameters produce good results on the validation set of that task $\mathcal{D}_{\mathcal{T}}^{\text{val}}$. The initialization of the parameters is trained by sampling minibatches of tasks from $\mathcal{D}$, computing the adapted parameters for all $\mathcal{D}_{\mathcal{T}}^{\text{train}}$ in the batch, evaluating the validation losses on the corresponding $\mathcal{D}_{\mathcal{T}}^{\text{val}}$ using the adapted parameters and finally backpropagating the sum of the validation losses through the adaptation step to the initial parameters $\theta$.

## 4 METHOD

We aim to develop a Multi-Modal Model-Agnostic Meta-Learner (MUMOMAML) that is able to quickly master a novel task sampled from *a multimodal task distribution*. To this end, we propose to leverage the ability of model-based meta-learners to identify tasks sampled from a task distribution as well as the ability of gradient-based meta-learners to consistently improve the performance with a few gradient steps. Specifically, we propose to learn a model-based meta-learner that produces a set of task-specific parameters to modulate the meta-learned prior parameters. Then, this modulated prior learns to adapt to a target task rapidly through gradient-based optimization. An illustration of our model is shown in Figure 1.

In this section, we first introduce our model-based meta-learner and a variety of modulation operators in section 4.1. Then we describe the training details for MUMOMAML in section 4.2 and present the implementation details in section 4.3.

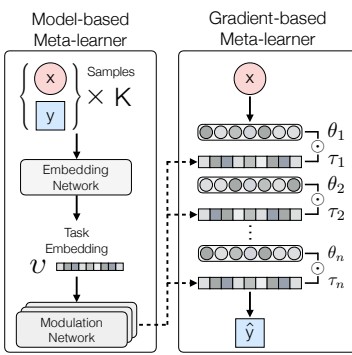

Figure 1: Overview of our model. Our model consists of two components: a model-based and gradient-based meta-learner. The former strives to identify the task from a few samples and modulate the prior accordingly; the latter performs gradient updates to effectively adapt to the specific task.

### 4.1 MODEL-BASED META-LEARNER

Given $K$ data points and labels $\{x_k, y_k\}_{k=1,\ldots,K}$, the task embedding network $f$ produces an embedding vector $\upsilon$ that encodes the characteristics of a task according to $\upsilon = f(\{x_k, y_k\}_{k=1,\ldots,K}; \omega_f)$. The modulation network $g$ modulates the meta-learned prior of the gradient-based meta-learner in the parameter space based on the task embedding vector $\upsilon$. To enable specialization of each block of the gradient-based meta-learner to the task, we apply the modulation block-wise to activate or deactivate the units of a block (*i.e.* a channel of a convolutional layer or a neuron of a fully-connected layer). Specifically, modulation network produces the modulation vectors for each block $i$ by $\tau_1, \ldots, \tau_N = g(\upsilon; \omega_g)$. We formalize the procedure of applying modulation as: $\phi_i = \theta_i \odot \tau_i$, where $\phi_i$ represents the modulated prior parameters for the gradient-based meta-learner, and $\odot$ represents a general modulation function. In the experiments, we investigate some representative modulation operations including attention-based modulation (Mnih et al., 2014; Vaswani et al., 2017) and feature-wise linear modulation (FiLM) (Perez et al., 2017).

**Attention based modulation** has been widely used in modern deep learning models and has proved its effectiveness across various tasks (Yang et al., 2016; Mnih et al., 2014; Zhang et al.,

2018; Xu et al., 2015). Inspired by the previous works, we employed attention to modulate the prior model. In concrete terms, attention over the outputs of all neurons (Softmax) or a binary gating value (Sigmoid) on each neuron's output is computed by the model-based meta-learner. These modulation vectors $\tau$ are then used to scale the pre-activation of each neural network layer $\mathbf{F}_\theta$, such that $\mathbf{F}_\phi = \mathbf{F}_\theta \otimes \tau$. Note that here $\otimes$ represents a channel-wise multiplication.

**Feature-wise linear modulation (FiLM)** has been proposed to modulate neural networks for achieving the conditioning effects of data from different modalities. We adopt FiLM as an option for modulating our gradient-based meta-learner. Specifically, the modulation vectors $\tau$ are divided into two components $\tau_\gamma$ and $\tau_\beta$ such that for a certain layer of the neural network with its pre-activation $\mathbf{F}_\theta$, we would have $\mathbf{F}_\phi = \mathbf{F}_\theta \otimes \tau_\gamma + \tau_\beta$. It can be viewed as a more generic form of attention mechanism. Please refer to Perez et al. (2017) for the complete details. Interestingly, in a recent few-shot learning paper FiLM modulation is used in a metric learning model and high performance on few-shot learning tasks is attained (Oreshkin et al., 2018).

## 4.2 Gradient-based Fast Adaptation

We perform a gradient-based meta-learning step to fine-tune the modulated initialization to fit the target task $\mathcal{T}_i$. The concrete training procedure is described by the following pseudo-code:

---
**Algorithm 1** Meta-Training Procedure for MuMoMAML.

---
**Input:** Task distribution $P(\mathcal{T})$, Hyper-parameters $\alpha$ and $\beta$
Randomly initialize $\theta$ and $\omega$.
**while** not DONE **do**
    Sample batches of tasks $\mathcal{T}_j \sim P(\mathcal{T})$
    **for** all j **do**
        Compute the modulation vector $\tau = g(\{x, y\}; \omega)$ with K samples from $\mathcal{D}_{\mathcal{T}_j}^{\text{train}}$
        Evaluate $\nabla_\theta \mathcal{L}_{\mathcal{T}_j}\left(f(x; \theta, \tau); \mathcal{D}_{\mathcal{T}_j}^{\text{train}}\right)$ with respect to the these K samples
        Compute adapted parameter with gradient descent: $\theta'_{\mathcal{T}_j} = \theta - \alpha \nabla_\theta \mathcal{L}_{\mathcal{T}_j}\left(f(x; \theta, \tau); \mathcal{D}_{\mathcal{T}_j}^{\text{train}}\right)$
    **end for**
    Update $\theta \leftarrow \theta - \beta \nabla_\theta \sum_{T_j \sim P(\mathcal{T})} \mathcal{L}_{\mathcal{T}_j}\left(f(x; \theta', \tau); \mathcal{D}_{\mathcal{T}_j}^{\text{val}}\right)$
    Update $\omega \leftarrow \omega - \beta \nabla_\omega \sum_{T_j \sim P(\mathcal{T})} \mathcal{L}_{\mathcal{T}_j}\left(f(x; \theta', \tau); \mathcal{D}_{\mathcal{T}_j}^{\text{val}}\right)$
**end while**

---

Note that the modulation vectors $\tau$ are not updated in the inner loop. The model-based learner is responsible for finding a good task-specific initialization through modulation. The gradient-based learning phase shown in the inner optimization loop is responsible for fitting the target task $\mathcal{T}$ with one or few gradient updates.

During the meta-testing phase, with a few data samples $\mathcal{D}_{\mathcal{T}_i}^{\text{train}}$ from task $\mathcal{T}_i$, we perform an adaptation step that corresponds to the inner loop of the meta-training procedure. Our model-based meta-learner first computes the task embedding based on the input data samples and then computes the modulation vector $\tau$ to modulate the prior model. Then, gradient updates are performed on the prior model $f(x; \theta, \tau)$ to compute the adapted parameters $\theta'$. Finally, we evaluate our adapted model $f(x; \theta', \tau)$ on the meta-test.

## 4.3 Implementation Details

For the model-based meta-learner, we used Seq2Seq (Sutskever et al., 2014) encoder structure to encode the sequence of $\{x, y\}_{k=1,...,K}$ with a bidirectional GRU (Chung et al.) and use the last hidden state of the recurrent model as the task embedding $\upsilon$. Then, the modulation vectors $\tau_i$ are computed from $\upsilon$ using a separate one hidden-layer multi-layer perceptron (MLP) for each layer of the gradient-based meta-learner. We experiment with our model in three representative learning scenarios namely regression, few-shot classification and reinforcement learning. The architectures used for each task differ from each other due to the differences in the task nature and data format. We discuss the tasks and the corresponding architectures in detail in section 5.

Table 1: Model's performance on the **multimodal 5-shot regression** with two or three modes. A Gaussian noise with $\mu = 0$ and $\sigma = 0.3$ is applied. The three modes regression is in general more difficult (thus higher error). In Multi-MAML, the GT modulation represents using ground-truth task identification to select different MAML models for each task mode. MUMOMAML (wt. FiLM) outperforms other methods by a margin.

| Configuration | | Two Modes (MSE) | | Three Modes (MSE) | |
|---|---|---|---|---|---|
| Method | Modulation | Post Modulation | Post Adaptation | Post Modulation | Post Adaptation |
| MAML (Finn et al., 2017) | - | 15.9255 | 1.0852 | 12.5994 | 1.1633 |
| Multi-MAML | GT | 16.2894 | 0.4330 | 12.3742 | 0.7791 |
| MUMOMAML (ours) | Softmax | 3.9140 | 0.4795 | 0.6889 | 0.4884 |
| MUMOMAML (ours) | Sigmoid | 1.4992 | 0.3414 | 2.4047 | 0.4414 |
| MUMOMAML (ours) | FiLM | 1.7094 | **0.3125** | 1.9234 | **0.4048** |

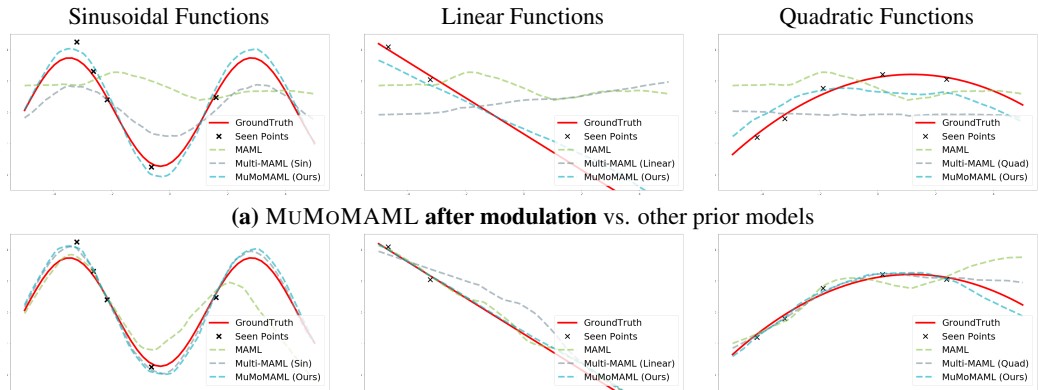

(a) MUMOMAML **after modulation** vs. other prior models

(b) MUMOMAML **after adaptation** vs. other posterior models

Figure 2: Qualitative Visualization of Regression on Three-modes Simple Functions Dataset. **(a)**: We compare the predicted function shapes of modulated MUMOMAML against the prior models of MAML and Multi-MAML, before gradient updates. Our model can fit the target function with limited observations and no gradient updates. **(b)**: The predicted function shapes after five steps of gradient updates, MUMOMAML is qualitatively better. More visualizations in Supplementary Material.

## 5 EXPERIMENTS

In this section, we evaluate MUMOMAML in a variety of tasks including regression, few-shot image classification and reinforcement learning. We design experiments with multimodal task distributions for regression and reinforcement learning tasks. To shed some light on the behavior of the proposed model on task distributions whose structure is not clearly multimodal or unimodal, we experiment with few-shot image classification.

### 5.1 REGRESSION

We investigate the capability of our model to learn few-shot regression tasks sampled from multimodal task distributions. In these tasks, a few input/output pairs $\{x_k, y_k\}_{k=1,...,K}$ sampled from a one dimensional function are given and the model is asked to predict $L$ output values $y_1^q, ..., y_L^q$ for input queries $x_1^q, ..., x_L^q$. We set up two regression settings with two task modes (sinusoidal and linear functions) or three modes (quadratic functions added).

As a baseline beside MAML, we propose Multi-MAML, which consists of $M$ (the number of modes) separate MAML models which are chosen for each query based on ground-truth task-mode labels. This baseline serves as an upper-bound for the performance of MAML when the task-mode labels are available.

The quantitative results are shown in Table 1. We observe that Multi-MAML outperforms MAML, showing that the performance of MAML degrades on multimodal task distributions. The marginal gap between the performance of our model in two and three mode settings indicates that MU-MOMAML is able to clearly identify the task modes and has sufficient capacity for all modes. Also, MUMOMAML consistently achieves better results than Multi-MAML demonstrating that

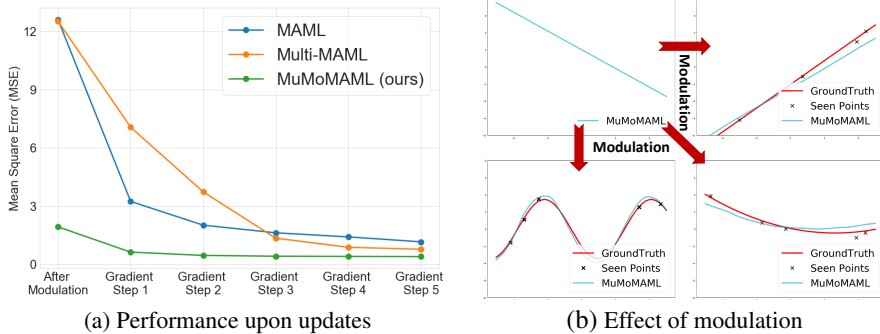

(a) Performance upon updates       (b) Effect of modulation

Figure 3: **(a)** Comparing the models' performance with respect to the number of gradient updates applied. For MUMOMAML, we report the performance after modulation for gradient step 0. **(b)** A demonstration of the modulation on prior model by our model-based meta-learner. With the FiLM modulation, MUMOMAML can adapt to different priors before gradient-based adaptation.

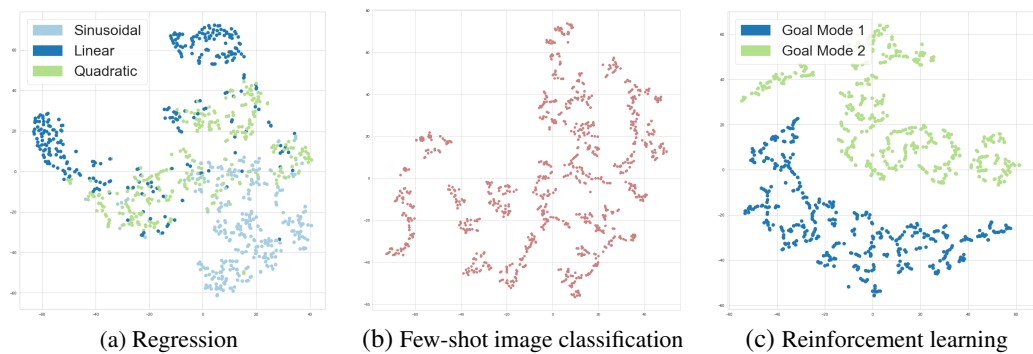

(a) Regression      (b) Few-shot image classification      (c) Reinforcement learning

Figure 4: tSNE plots of the task embeddings produced by our model from randomly sampled tasks; marker color indicates different modes of a task distribution. The plots (a) and (c) reveal a clear clustering according to different task modes, which demonstrates that MUMOMAML is able to identify the task from a few samples and produce a meaningful embedding $\upsilon$. (a) Regression: the distance among distributions aligns with the intuition of the similarity of functions (*e.g.* a quadratic function can sometimes be similar to a sinusoidal or a linear function while a sinusoidal function is usually different from a linear function) (b) Few-shot image classification: we observe a embedding manifold with some sub-structures appearing. However, it is not intuitive to understand them directly. (c) Reinforcement learning: the embeddings for 2D navigation goals sampled from two Gaussian distributions environment are cleanly separated.

our model is able to discover and exploit transferable knowledge across different modes to improve its performance.

The qualitative results are shown in Figure 2, visualizing the predicted functions. We observe that MUMOMAML is able to effectively modulate its meta-learned prior to fit a sampled task (see Figure 3 (a)), which greatly eases the optimization procedure of our gradient-based learner (see Figure 3 (b)). To gain some insights into what kind of task embeddings are produced by our model, we show tSNE (Maaten & Hinton, 2008) visualization of the predicted embedding vectors $\upsilon$ in Figure 4 (a). The tSNE plot shows that our model is able to capture the mode structure in the embeddings, enabling performance gain from modulation.

We compared attention modulation with Sigmoid or Softmax and FiLM modulation and found that FiLM achieves better results. We therefore use FiLM in the subsequent experiments. Please refer to Supplementary Material for additional results.

## 5.2 FEW-SHOT IMAGE CLASSIFICATION

The task of few-shot image classification considers a problem of classifying images into N classes with a small number (K) of labeled samples available. To evaluate our model on this task, we

conduct experiments on OMNIGLOT, a widely used handwritten character dataset of binary images. The results are shown in Table 2, demonstrating that our method achieves comparable or better results against state-of-the-art algorithms. Please refer to Supplementary Material for details.

To gain insights to the task embeddings $\upsilon$ produced by our model, we again randomly sampled 2000 tasks and employ tSNE to visualize the $\upsilon$ in Figure 4 (b). While we are not able to clearly distinguish the modes of task distributions, we observe that the distribution of the produced embeddings is not uniformly distributed or unimodal, potentially indicating the model-based meta-learner finding some exploitable structure in this task distribution.

Table 2: 5-way and 20-way, 1-shot and 5-shot classification accuracy on OMNIGLOT Dataset. For each task, the best-performing method is highlighted. MUMOMAML achieves comparable or better results against state-of-the-art few-shot learning algorithms for image classification.

| Method | OMNIGLOT | | | |
| | 5 Way Accuracy (in %) | | 20 Way Accuracy (in %) | |
| | 1-shot | 5-shot | 1-shot | 5-shot |
| --- | --- | --- | --- | --- |
| Siamese nets Koch et al. (2015) | 97.3 | 98.4 | 88.2 | 97.0 |
| Matching nets Vinyals et al. (2016) | 98.1 | 98.9 | 93.8 | 98.5 |
| Meta-SGD Li et al. (2017) | 99.5 | **99.9** | 95.9 | 99.0 |
| Prototypical nets Snell et al. (2017) | 97.4 | 99.3 | 96.0 | 98.9 |
| SNAIL Mishra et al. (2018) | 99.1 | 99.8 | **97.6** | **99.4** |
| T-net Lee & Choi (2018) | 99.4 | - | 96.1 | - |
| MT-net Lee & Choi (2018) | 99.5 | - | 96.2 | - |
| MAML (Finn et al., 2017) | 98.7 | **99.9** | 95.8 | 98.9 |
| MUMOMAML (ours) | **99.7** | **99.9** | 97.2 | **99.4** |

## 5.3 REINFORCEMENT LEARNING

We experiment with MUMOMAML in reinforcement learning (RL) to verify its ability to learn to rapidly adapt to tasks sampled from multimodal task distributions given a minimum amount of interaction with an environment. Specifically, we consider a Markov decision process (MDP) with horizon $H$ formed by a distribution of tasks $T_i$. Each task has an initial state distribution $p_i(s)$, a transition distribution $p_i(s_{t+1}|s_t, a_t)$, and a reward function $R_i(s_t, a_t)$. Each rollout contains $(s_1, a_1, ..., s_H)$ and its corresponding rewards $R(s_t, a_t)$ and $K$ rollouts are available for $K$-shot RL.

To analyze our model's performance on an MDP formed by a multimodal distribution of tasks, we designed and conducted experiments on a 2D navigation environment similar to Finn et al. (2017). In the environment, an agent is trained to navigate to a goal location sampled from a bimodal distribution. The agent takes its current location as input and has an action space limited to a 2D vector representing the moving orientation. At each time step, it receives a reward defined as the negative distance to the goal. Instead of uniformly sampling goals from a square like Finn et al. (2017), we sample goals from $M$ Gaussian distributions that are far away from each others, which makes our environment suitable for our analysis but more challenging.

To identify the mode of a task distribution in reinforcement learning, we run the meta-learned prior model without modulation or adaptation to interact with the environment and collect a single trajectory and obtained rewards. Then the collected trajectory and rewards are fed to our model-based meta-learner to compute the task embedding $\upsilon$ and the modulation vectors $\tau$.

Next, the gradient-based meta-learner operates on the modulated policy and seeks to rapidly adapt to the environment – approach the goal. We refer the reader to A.3 for more details.

We compare MUMOMAML against MAML. In all experiments, we set $H = 100$, and $M = 2$. The average returns are shown in Figure 5 (a), demonstrating that our model is able to achieve a significant performance gain from modulation and can consistently outperform MAML. Qualitative comparisons are shown in Figure 6. After taking a few gradient updates, both the models are able to reach to goals while MUMOMAML converges in fewer steps. Please refer to Supplementary Material for additional results.

To investigate the correlation between sampled goals and produced embeddings $\upsilon$, we sample 1000 goals, collect the trajectories generated by the prior model and compute the task embeddings for the goals. As shown in Figure 4 (c), a clear separation of goals sampled from two goal modes indicates

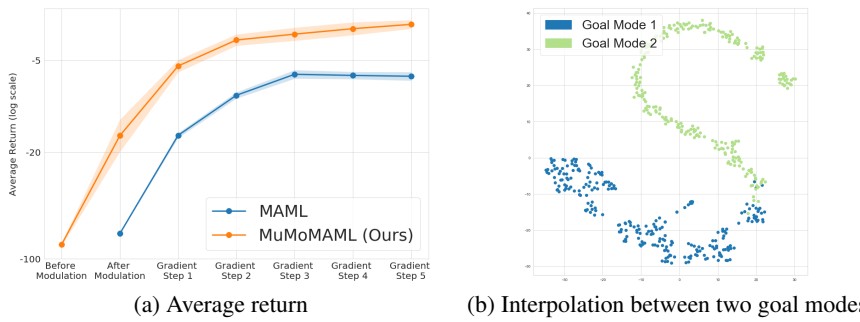

(a) Average return        (b) Interpolation between two goal modes

Figure 5: (a) A comparison between MUMOMAML (orange) and MAML (blue) on a 2D navigation task. For MUMOMAML the **Before Modulation** return corresponds to the return of the first trajectory sampled for computing the task embedding. (b) A tSNE plot of task embeddings of interpolated goals between the two centers of the task modes in the 2D navigation task.

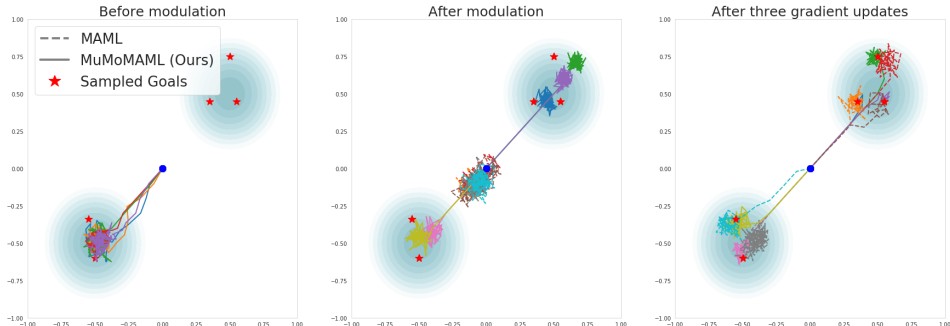

Figure 6: Sample trajectories in the 2D navigation environment. In the before modulation plot on the left, the MUMOMAML policy (line) moves randomly around one of the modes. In the middle plot, before gradient updates are applied, MUMOMAML navigates rapidly to the correct modes, while MAML policy (dashed line) has not yet observed any rewards. On the right, after three gradient updates, both have found good policies for the goals, while MUMOMAML converges on the goals in fewer steps.

that our meta-learned initial policy provides useful trajectories and our model-based learner is able to produce meaningful embeddings. To further examine this hypothesis, we uniformly sample goals along a straight line from the center of a goal mode to the center of the other goal mode and compute the embeddings. As shown in Figure 5, the clean curved structure demonstrate the competence of our model-based meta-learner.

## 6 CONCLUSION

We present a novel approach that is able to leverage the strengths of both model-based and gradient-based meta-learners to discover and exploit the structure of multimodal task distributions. Given a few samples from a target task, our model-based learner first identifies the mode of the task distribution and then modulates the meta-learned prior in a parameter space. Next, the gradient based meta-learner efficiently adapts to the target task through gradient updates. We empirically observe that our model-based learner is capable of effectively recognizing the task modes and producing embeddings that captures the structure of a multimodal task distribution. We evaluated our proposed model in few-shot regression, image classification and reinforcement learning, and achieved superior generalization performance on tasks sampled from multimodal task distributions.

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

# A    ADDITIONAL EXPERIMENTAL DETAILS

## A.1    REGRESSION

**Setups.**    To form multimodal task distributions, we consider a family of functions including sinusoidal functions (in forms of $A \cdot \sin w \cdot x + b + \epsilon$, with $A \in [0.1, 5.0]$, $w \in [0.5, 2.0]$ and $b \in [0, 2\pi]$), linear functions (in forms of $A \cdot x + b$, with $A \in [-3, 3]$ and $b \in [-3, 3]$) and quadratic functions (in forms of $A \cdot (x - c)^2 + b$, with $A \in [-0.15, -0.02] \cup [0.02, 0.15]$, $c \in [-3.0, 3.0]$ and $b \in [-3.0, 3.0]$ ). A Gaussian observation noise with $\mu = 0$ and $\epsilon = 0.3$ is added to each data point sampled from the target task. In all the experiments, $K$ is set to 5 and $L$ is set to 10. We report the mean squared error (MSE) as the evaluation criterion. Due to the multimodality and uncertainty, this setting is more challenging comparing to (Finn et al., 2017).

**Models and Optimization.**    In the regression task, we trained a 4-layer fully connected neural networks with the hidden dimensions of 100 and ReLU non-linearity for each layer, as the base model for both MAML and MUMOMAML. In MUMOMAML, an additional model with a Bidirectional GRU of hidden size 40 associated with multiple linear layers is trained to generate $\tau$ and to modulate each layer of the base model. We used the same hyper-parameter settings as the regression experiments presented in Finn et al. (2017) and used Adam Kingma & Ba (2015) as the meta-optimizer. For all our models, we train on 5 meta-train examples and evaluate on 10 meta-val examples to compute the loss.

## A.2    FEW-SHOT IMAGE CLASSIFICATION

**Setups.**    In the few-shot learning experiments, we used OMNIGLOT, a dataset consists of 50 languages, with a total of 1632 different classes with 20 instances per class. Following Santoro et al. (2016), we downsampled the images to $28 \times 28$ and perform data augmentation by rotating each member of an existing class by a multiple of 90 degrees to form new data points of a given class.

**Models and Optimization.**    Following prior works (Vinyals et al., 2016; Finn et al., 2017), we used the same 4-layer convolutional neural network and applied the same training and testing splits from Finn et al. (2017) and compare our model against baselines for 5-way and 20-way, 1-shot and 5-shot classification.

## A.3    REINFORCEMENT LEARNING

**Environment.**    In the bimodal 2D navigation environment the goals are sampled from one of two bivariate Gaussians with means of $[0.5, 0.5]$ and $[-0.5, -0.5]$ and standard deviation of 0.5. Each mode is selected with equal probability. In the beginning of each episode, the agent starts from the origin. The agent's observation is its 2D-location and the reward is the negative squared distance to the goal. The agent does not observe the goal directly, instead it must learn to navigate there based on the reward function alone. The episodes terminate after 100 steps or when the agent comes to the distance of 0.01 from the goal.

**Models and Optimization.**    The first trajectory from the environment is sampled using the unmodulated, unadapted model and used for computing the task embedding. We experimented with sampling more trajectories for this purpose, but found no improvement over using only one. The batch of trajectories used for computing the first gradient-based adaptation step is sampled using the modulated model and the batches after that use the modulated and adapted parameters from the previous update step. For the gradient adaptation steps, we use the vanilla policy gradient algorithm Williams (1992). As the meta-optimizer we use the trust region policy optimization algorithm Schulman et al. (2015). With respect to the gradient-based adaptation we follow the meta-learning procedure described in Finn et al. (2017).

The models are trained for one gradient adaptation step with a batch size of 20 trajectories. We use 20 tasks for the meta-batch. We chose the hyperparameters of MUMOMAML and MAML through random search. For MAML we use the inner loop update step size 0.03 and the discounting parameter $\gamma$ of 0.95, for MUMOMAML we use step size 0.01 and $\gamma$ 0.99. MAML uses a two-layer MLP model with hidden size 32 and ReLU activations. For MUMOMAML, we use the same size

MLP in addition to the model-based meta-learner, which consists of an RNN for the embedding network and an MLP for the modulation network. The RNN model used is the GRU with hidden size 8. The number of hidden units in the embedding network is 8.

## B  ADDITIONAL EXPERIMENTAL RESULTS

For the better understanding of our paper, we provide additional results in this section.

### B.1  ADDITIONAL QUALITATIVE RESULTS FOR REGRESSION

Additional qualitative results for MUMOMAML after modulation are shown in Figure 7 and additional qualitative results for MUMOMAML after adaptation are shown in Figure 8.

### B.2  ADDITIONAL QUALITATIVE RESULTS FOR REINFORCEMENT LEARNING

Additional trajectories sampled from the 2D navigation environment are presented in Figure 9.

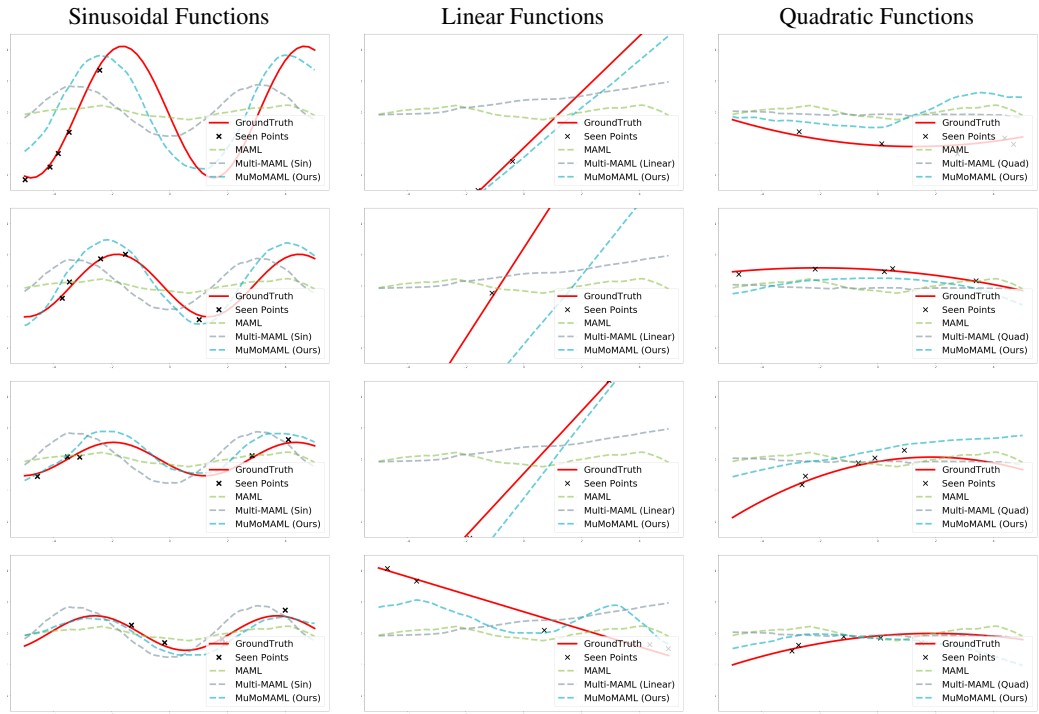

Figure 7: Additional qualitative results of the regression tasks **(a)**: MUMOMAML **after modulation** vs. other prior models.

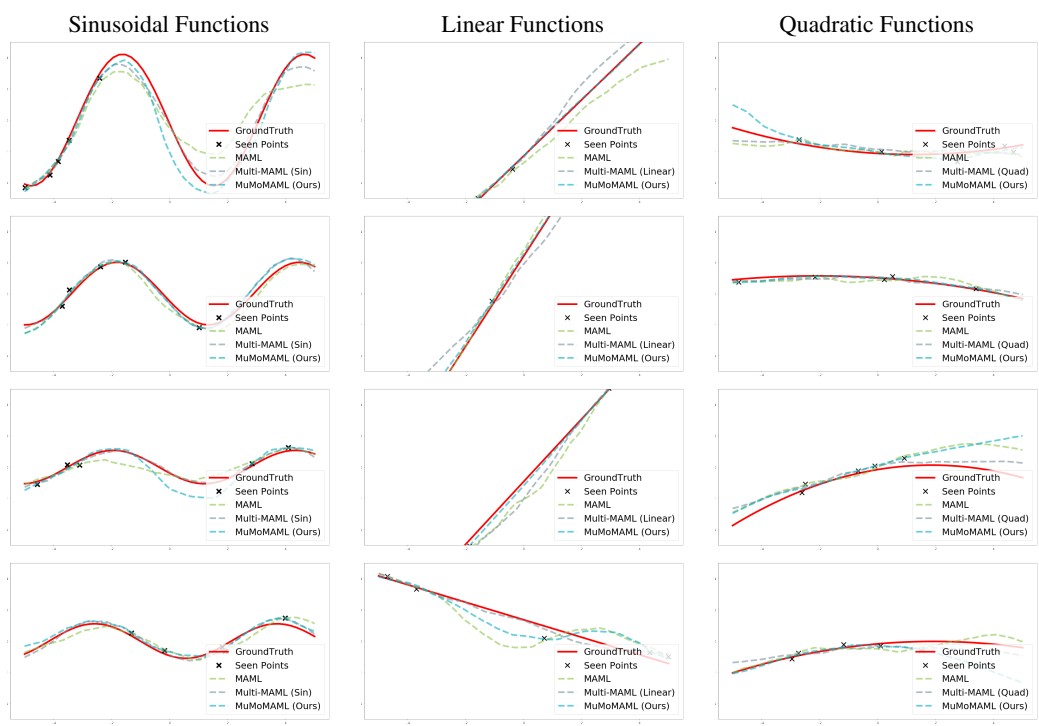

Figure 8: Additional qualitative results of the regression tasks **(b)**: MUMOMAML **after adaptation** vs. other posterior models.

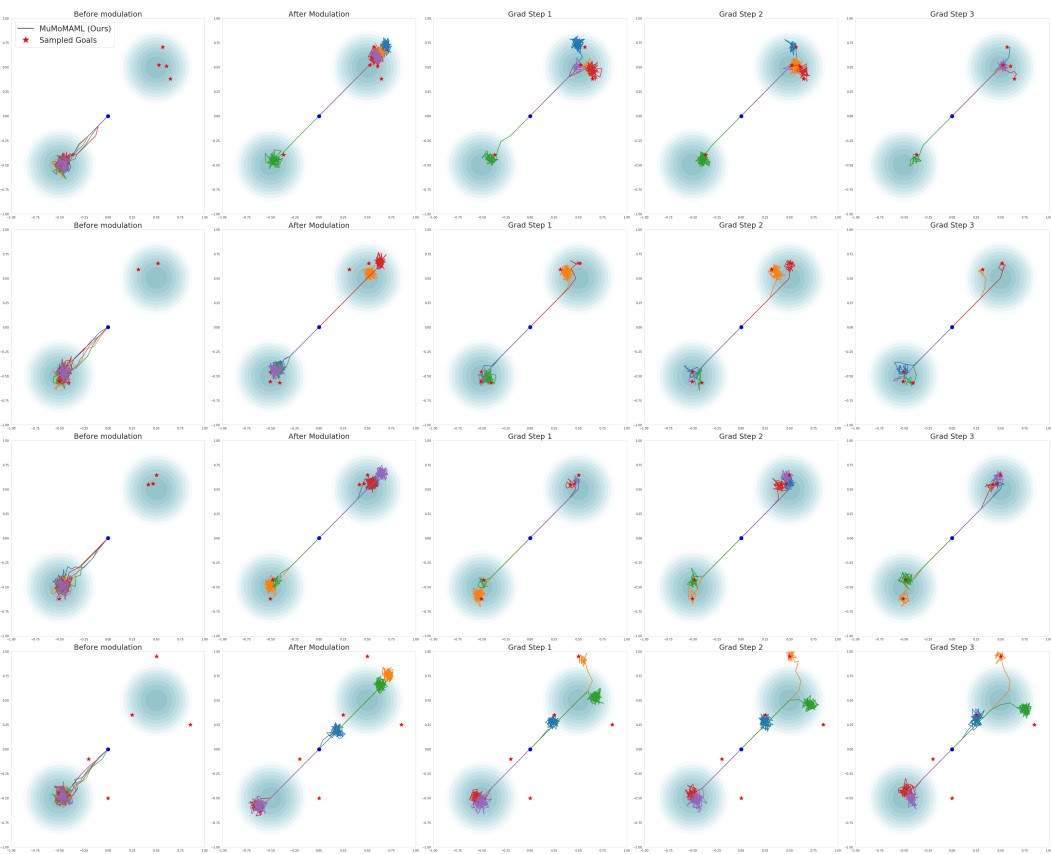

Figure 9: Additional trajectories sampled from the 2D navigation environment with MuMoMAML. The first four rows are with goals sampled from the environment distribution, where MuMoMAML demonstrates rapid adaptation and is often able to locate the goal exactly. On the fifth row, trajectories are sampled with extrapolated goals. The agent is left farther away from the extrapolated goals after the modulation step, but the gradient based adaptation steps then steadily recover the performance.

