# OpenReview forum: "Model-Agnostic Meta-Learning for Multimodal Task Distributions"
_ICLR.cc/2019/Conference_

### Official Review · AnonReviewer2 · 2018-11-02

**Rating:** 5
**Confidence:** 3

**Review:**


MuMoMAML: Model-Agnostic Meta-Learning for Multimodal Task Distributions

This paper proposed multi-modal MAML, which alleviates the single initialization limitation of MAML by modulating task prior with MAML. Below are some comments.

Pros:
1. Overall, the paper is clear written.
2. By using modulation, there is no need to explicitly control/know the number of modes in advance.
3. The multi-MAML baseline is good for an ablation study, though it is only on a synthetic regression task.
4. MUMOMAML combines the strength of both gradient-based and model-based meta-learners.

Cons.
1. The novelty of the paper seems to be the combinations of MAML and FiLM, which seems a bit limited.
2. I wonder whether the proposed method is mostly useful when there is a clear mode difference as in the synthetic regression/RL tasks of the paper. Moreover, the paper only shows tasks with only two-three modes, what happen when there is a large number of modes?
3. What's the results on the mini-Imagenet? The Omniglot seems to be saturated already.
4. Why tau is not updated in the inner loop of Algorithm 1?

Minor:
1. page 4, 'in to' -> 'into'
2. In page 5, in 'based on the input data samples and then
infers the parameter to modulate the prior model', what does the `input data samples' refers to? Is it the training data of a meta-learning task?
3. Do you stop gradient to the learner in MUMOMAML?

---

> ### Author Response · Authors · 2018-11-20
> **Response to AnonReviewer2**
>
>
> We thank the reviewer for the feedback and address the concerns in detail below.
>
> > The novelty of the paper seems to be the combinations of MAML and FiLM, which seems a bit limited.
>
> We propose a previously under-explored problem of enabling a family of meta-learners that seek a parameter initialization [1-6] to deal with multimodal task distributions. Also, we empirically demonstrate a limitation of a state-of-the-art gradient-based meta-learners (MAML) - it is impractical to find a single initialization that allows a network to adapt to diverse tasks with few gradient steps, which is also pointed out by R3. We then present a framework together with a training algorithm that aims to alleviate the problem. We propose to identify the mode of a task and activate or deactivate some parts of a network to enable efficient adaptation with few gradient steps. We experiment with different modulation methods (e.g. attention mechanisms with sigmoid and softmax) and show promising results in section 5.1, which is not limited to FiLM.
>
> > I wonder whether the proposed method is mostly useful when there is a clear mode difference as in the synthetic regression/RL tasks of the paper.
>
> The results presented in the paper demonstrate that our model can deal with overlapping modes. Specifically, in the regression experiments reported in the paper, the parameter ranges of the functions in the dataset allow for generation of functions that are indistinguishable from functions of a different class. That is, a sinusoid with a low enough frequency will look like a linear function in the given coordinate range, especially considering that noise is added to the observation. Also, in 2D navigation environment, sampled goals can appear between the mode centers.
>
> > What's the results on the mini-Imagenet? The Omniglot seems to be saturated already.
>
> The fact that miniImageNet has a higher dimensional visual data (much larger image size) comparing to Omniglot makes it much more difficult to learn good task embeddings using our simple task embedding network (linear projection + BiGRU). We performed additional experiments on miniImageNet during the rebuttal and found that the few-shot learning result of our model is only on par with MAML. We believe that with a better tuned convolutional structure or pre-trained feature extractors (as a practice suggested in [1,2]) for learning task embedding from high-dimensional visual data, our model could potentially improve.
>
> > Why tau is not updated in the inner loop of Algorithm 1?
>
> \tau is not updated in the outer or inner loop, because it is not a parameter of our model; instead, it is a set of vectors produced by our model-based meta-learner to activate or deactivate some parts of the gradient-based meta-learner accordingly to the estimated task mode. Therefore, only the parameters of the gradient-based meta-learner (\theta) are updated in the inner loop training. We revised the paper to make this clear.
>
> > In page 5, in 'based on the input data samples and then infers the parameter to modulate the prior model', what does the `input data samples' refers to? Is it the training data of a meta-learning task?
>
> Yes. Input data samples (x_1, y_1, …, x_K, y_K) form a meta-learning task. We revised the paper to make this clear.
>
> > Do you stop gradient to the learner in MUMOMAML?
>
> The gradients of the loss for a single task, that is computed in the inner loop of the algorithm, are only used for adapting the parameters of the gradient-based meta-learner. So in this sense, the inner loop gradients are stopped to the model-based meta-learner. The outer loop loss is used to compute gradients with respect to the initial parameters of the gradient-based meta-learner and the parameters of the model-based meta-learner.
>
> In [1] they experiment with the “first-order MAML” by stopping the gradient through the inner loop update procedure when updating the initial parameters. As [1] does not report a significant difference, we do not do this.
>
> > page 4, 'in to' -> 'into'
>
> We appreciate the reviewer pointing this out. We revised the paper accordingly.
>
> [1] Finn et al. “Model-Agnostic Meta-Learning for Fast Adaptation of Deep Networks”, ICML 2017
> [2] Finn et al. ”Probabilistic Model-Agnostic Meta-Learning”, NIPS 2018
> [3] Kim et al. “Bayesian Model-Agnostic Meta-Learning”, NIPS 2018
> [4] Lee and Choi “Gradient-Based Meta-Learning with Learned Layerwise Metric and Subspace”, ICML 2018
> [5] Grant et al. “Recasting Gradient Based Meta-Learning as Hierarchical Bayes”, ICLR 2018
> [6] Nichol et al. “Reptile: a Scalable Meta-learning Algorithm”, arXiv 2018
> [7] Oreshkin et al. “Task dependent adaptive metric for improved few-shot learning”, NIPS 2018
> [8] Qiao et al. “Few-Shot Image Recognition by Predicting Parameters from Activations”, CVPR 2018

---

### Official Review · AnonReviewer3 · 2018-11-04
**Why modulation works for meta-learning**

**Rating:** 5
**Confidence:** 4

**Review:**

This paper presents an interesting meta-learning algorithm that can learn from multimodal task distributions, by combining model-based and gradient-based meta-learning. It first represents a task with a latent feature vector produced by a recurrent network, and then modulates the meta-learned prior with this task-specific latent feature vector before applying gradient-based adaptation. Experimental results are shown to validate the proposed algorithm. While the idea appears to be quite novel for meta-learning, further efforts are needed to improve this work.

1. The experiment on few-shot image classification is less convincing, with results only on the Omniglot dataset, which are only comparable to those of existing methods that are designed for a single task distribution. Why not show results on MiniImageNet or other more realistic datasets which are more likely to be multimodal?

2. It is not clear how the idea of modulation works for multimodal meta-learning. More discussions and insights can be helpful.

3. The encoding of a task relies on the order of examples, which seems undesirable for a classification or regression problem.

---

> ### Author Response · Authors · 2018-11-20
> **Response to AnonReviewer3**
>
>
> We thank the reviewer for the feedback and address the concerns in detail below.
>
> > The experiment on few-shot image classification is less convincing, with results only on the Omniglot dataset, which are only comparable to those of existing methods that are designed for a single task distribution. Why not show results on MiniImageNet or other more realistic datasets which are more likely to be multimodal?
>
> The fact that miniImageNet has a higher dimensional visual data (much larger image size) comparing to Omniglot makes it much more difficult to learn good task embeddings using our simple task embedding network (linear projection + BiGRU). We performed additional experiments on miniImageNet during the rebuttal and found that the few-shot learning result of our model is only on par with MAML. We believe that with a better tuned convolutional structure or pre-trained feature extractors (as a practice suggested in [1,2]) for learning task embedding from high-dimensional visual data, our model could potentially improve.
>
> > It is not clear how the idea of modulation works for multimodal meta-learning. More discussions and insights can be helpful.
>
> Our intuition is that each block (i.e. a channel of a convolutional layer or a neuron of a fully-connected layer) of the gradient-based meta-learner network should learn to be specialized in different meta-learning tasks. To utilize this specialization to ensemble a powerful meta-learner, we apply the modulation block-wise to activate or deactivate the units of a block by estimating the mode of the task using the model-based meta-learner. In other words, we propose to first select a relevant subnetwork based on a given task to enable efficient adaptation. We revised the paper to make this intuition clear.
>
> > The encoding of a task relies on the order of examples, which seems undesirable for a classification or regression problem.
>
> In fact, the bidirectional GRU (BiGRU) in our model can be considered as a superset of functions that are order-invariant. According to [1], an order-invariant function that takes a set/bag of data would have the following forms:
> (1) A transformation applied to each instance data to obtain instance embedding
> (2) Another different transformation applied to the sum of those instance embeddings and get a holistic embedding represents all data in the set.
> In this case, our BiGRU first embeds instances into a feature representation. Though there is a dependency to previous and later instances, it could be set to zero through the learning and therefore result in an instance-wise embedding function like (1). Next, we average the output of BiGRU and perform an additional linear transformation to compute \tau, which is the same as (2).
> Therefore, we would like to emphasize that through using BiGRU, our model-based meta-learner can be order-invariant or order-dependent based on the underlying structure inside the training data, which has placed more flexibility. The detailed discussion is included in the revised paper.
>
> [1] Oreshkin et al. “Task dependent adaptive metric for improved few-shot learning”, NIPS 2018
> [2] Qiao et al. “Few-Shot Image Recognition by Predicting Parameters from Activations”, CVPR 2018
> [3] Zaheer et al. “Deep Sets”, NIPS 2017

---

### Official Review · AnonReviewer1 · 2018-11-06
**Layer-wise conditioning via task-embedding for meta-learning**

**Rating:** 3
**Confidence:** 5

**Review:**

Strengths:
+ The paper identifies a valid limitation of the MAML algorithm: With a limited number of gradient descent steps from a single initialization, there is a limit to the ability of a fixed-size neural network to adapt to tasks sampled from a diverse dataset.
+ The tSNE plots show some preliminary interesting structure for the simple regression and RL tasks, but not for the classification task.

Weaknesses:
- The motivation of uncovering latent modes of a task distribution does not align with the proposed method. The algorithm computes a continuous representation of the data from a task (which is fixed during gradient-based fast adaptation). The mode identity, on the other hand, should be a discrete variable.  Such a discrete variable is never explicitly computed in the proposed method.
- The technical writing is unclear and jargon is often used without definition. Importantly, one of the central motivators of the paper, "task modulation", is never given a precise definition.
- The standard few-shot classification task (Omniglot) does not clearly consist of a task distribution that is multimodal, so the method is not well-motivated in this setting.
- Experimental conclusions are weak.

Major comments:
- The paper neglects to discuss how the proposed method could be used in the context of other methods for "gradient-based meta-learning" such as Ravi & Larochelle (2016). I believe the attention-based modulation and the FiLM modulation could be easily adapted to that setting. Why was this not discussed or evaluated?
- Conditioning has been used in the context of few-shot learning before, but this is not discussed (https://arxiv.org/abs/1805.10123, https://arxiv.org/abs/1806.07528).
- The paper often confounds task representation with neural network parameter values. For example, Figure 1 depicts the adaptation of parameter values with gradients (\nabla L), yet the caption describes "task modes." More careful writing would disentangle these two components.
- The motivation for the particular form of the task embedding computation is not given. What were the other options? Why not, for example, an order-invariant function instead of a bidirectional GRU?
- In all of the experiments, there is no appropriate baseline that keeps the parameter dimensionality constant, so it is unclear whether the (marginal) improvement in performance is due to added expressivity by adding more parameters rather than an algorithmic improvement. I suggest an ensembling baseline with an appropriate number of ensemble members.
- There is no evaluation on a standard benchmark for few-shot classification (miniImageNet), and the Omniglot improvement is small.
- The reinforcement learning comparison at some point compares MUMOMAML with modulation applied (therefore with access to task-specific data) to MAML with no adaptation (and therefore no access to task-specific data). This is not entirely fair.
- tSNE results can be misleading (e.g., see https://distill.pub/2016/misread-tsne/), and the task delineation is not extremely clean. I would be more convinced if a clustering algorithm were applied.

Minor comments:
- The paper needs to be checked over for English grammar and style.
- everywhere: The "prior" referred to in this paper is not a prior in the Bayesian sense. I suggest a more careful use of terminology.
- abstract: "augment existing gradient-based meta-learners" You augment a specific variant of gradient-based meta-learning, MAML.
- pg. 1: "carve on a snowboard" don't know what this means
- The terminology of "task distribution" and "modes" thereof is used without introduction in the introduction section. The terminology "model-based meta-learning/adaptation" and "gradient-based meta-learning/adaptation" is also used without introduction here. This makes the introduction unnecessarily opaque. Consider the reader who is not familiar with meta-learning papers; they would have a very hard time parsing, for example, the phrase "...this not only requires additional identity information about the modes, which is not always available or is ambiguous when the modes are not clearly disjoint..." (pg. 1).
- Further, the terminology "model-based" seems non-standard, and is aliased with the term model-based reinforcement learning (which specifically refers to the set of RL algorithms that make use of a "model" of transition dynamics). Since the paper tackles a reinforcement learning benchmark, this may lead to some confusion.
- pg. 3; "our model does not maintain an internal state" Is the task representation/embedding not an internal state?
- pg. 3: "relevant but vaguely related skills" this is imprecise
- pg. 3: The episodic training setup, which is standard to meta-learning setups, could be much better described. The MAML algorithm could be given better intuition.
- everywhere: "task specific" -> task-specific
- Algorithm 1: "infer" is a misuse of terminology that usually refers to an operation in latent variable probabilistic modelling. Since the computation of \tau is purely feedforward, I recommend writing "compute."
- \tau should be used in some places where v is used instead

---

> ### Author Response · Authors · 2018-11-20
> **Response to AnonReviewer1 (Part 1/3)**
>
> We thank the reviewer for the feedback and address the concerns in detail below.
>
> > The motivation of uncovering latent modes of a task distribution does not align with the proposed method. The algorithm computes a continuous representation of the data from a task (which is fixed during gradient-based fast adaptation). The mode identity, on the other hand, should be a discrete variable. Such a discrete variable is never explicitly computed in the proposed method.
>
> The reason we selected a continuous representation for the task identity vector is that we believe many task distributions of interest have richer structure than a distribution with clearly disjoint modes. For example, the task distribution studied in the regression experiment has multimodal structure, but the functions are parameterized in such way that the sampled functions interpolate the modes densely (i.e. the observed data points of a quadratic function can sometimes be very similar to a sinusoidal function, as shown in Figure 7 and 8 in the paper). We believe it is a strength of our proposed method that the identity vector can help interpolate modes in a task space where some tasks fall in between the modes, as pointed out by R2. We revised the paper to better reflect this idea so that discrete mode identification is not implied.
>
> > The technical writing is unclear and jargon is often used without definition. Importantly, one of the central motivators of the paper, "task modulation", is never given a precise definition.
>
> We revised the Introduction, Preliminaries and Method sections to improve the language and make them easier to follow. The term “task modulation” is not mentioned in the original paper. To address the reviewer’s concern, we provide a clear description of the term “modulation” where it is mentioned.
>
> > The standard few-shot classification task (Omniglot) does not clearly consist of a task distribution that is multimodal, so the method is not well-motivated in this setting.
>
> The value of comparing the proposed model against baselines on few-shot classification image is to verify if the proposed model can achieve better performance when the task modes are not obvious. This intuition is stated in section 5 in the original paper.
>
> > The paper neglects to discuss how the proposed method could be used in the context of other methods for "gradient-
> based meta-learning" such as Ravi & Larochelle (2016). I believe the attention-based modulation and the FiLM modulation could be easily adapted to that setting. Why was this not discussed or evaluated?
>
> We focus on improving the family of gradient-based meta-learners which seek a common initialization for the parameters to enable fast adaptation within few gradient steps, including [1-6]. While Ravi & Larochelle (2016), which is included the original paper under the category of optimization-based meta-learner, propose to update the meta-learner with gradients by learning update rules (as mentioned in section 2), it does not seek a parameter initialization. As a result, it does not suffer from the issue that we aim to solve here, and therefore we do not see an obvious way to augment this work with our proposed model.
>
> > Conditioning has been used in the context of few-shot learning before, but this is not discussed (https://arxiv.org/abs/1805.10123, https://arxiv.org/abs/1806.07528).
>
> We appreciate the suggestion and we have included the references in the revised paper.
>
> > The paper often confounds task representation with neural network parameter values. For example, Figure 1 depicts the adaptation of parameter values with gradients (\nabla L), yet the caption describes "task modes." More careful writing would disentangle these two components.
>
> We determined that Figure 1 (a) is perhaps not fulfilling its purpose in providing an approachable visual overview of the method and removed it to avoid further confusion. We revised writing in the paper to better separate the concepts of parameters and task representation. Specifically, we changed the language so that \tau is not called a parameter vector, but rather a “modulation vector” to reflect the fact that it is computed by the model.

---

> ### Author Response · Authors · 2018-11-20
> **Response to AnonReviewer1 (Part 2/3)**
>
>
> > The motivation for the particular form of the task embedding computation is not given. What were the other options? Why not, for example, an order-invariant function instead of a bidirectional GRU?
>
> In fact, the bidirectional GRU (BiGRU) in our model can be considered as a superset of functions that are order-invariant. According to [7], an order-invariant function that takes a set/bag of data would have the following forms:
> (1) A transformation applied to each instance data to obtain instance embedding
> (2) Another different transformation applied to the sum of those instance embeddings and get a holistic embedding represents all data in the set.
> In this case, our BiGRU first embeds instances into a feature representation. Though there is a dependency to previous and later instances, it could be set to zero through the learning and therefore result in an instance-wise embedding function like (1). Next, we average the output of BiGRU and perform an additional linear transformation to compute \tau, which is the same as (2).
> Therefore, we would like to emphasize that through using BiGRU, our model-based meta-learner can be order-invariant or order-dependent based on the underlying structure inside the training data, which has placed more flexibility. The detailed discussion is included in the revised paper.
>
> > In all of the experiments, there is no appropriate baseline that keeps the parameter dimensionality constant, so it is unclear whether the (marginal) improvement in performance is due to added expressivity by adding more parameters rather than an algorithmic improvement. I suggest an ensembling baseline with an appropriate number of ensemble members.
>
> We experimented with adding layers and adding more units to the MAML layers, and it does not improve performance on the studied problems.
>
> > There is no evaluation on a standard benchmark for few-shot classification (miniImageNet), and the Omniglot improvement is small.
>
> The fact that miniImageNet has a higher dimensional visual data (much larger image size) comparing to Omniglot makes it much more difficult to learn good task embeddings using our simple task embedding network (linear projection + BiGRU). We performed additional experiments on miniImageNet during the rebuttal and found that the few-shot learning result of our model is only on par with MAML. We believe that with a better tuned convolutional structure for learning task embedding from high-dimensional visual data, our model could potentially improve.
>
> > The reinforcement learning comparison at some point compares MUMOMAML with modulation applied (therefore with access to task-specific data) to MAML with no adaptation (and therefore no access to task-specific data). This is not entirely fair.
>
> In the figures we show the modulated MuMoMAML and MAML at the same step to align the gradient update steps on both algorithms, but the intention was not to compare the modulated model to the unadapted MAML. We clarified this in the caption and the text.
>
> > tSNE results can be misleading (e.g., see https://distill.pub/2016/misread-tsne/), and the task delineation is not extremely clean.
>
> We would like to emphasize that the structure shown in our embedding visualization is consistent across a range of perplexity choices. In fact, although the distill article argues that "random embedding can have some non-random sub-cluster structure", it is hard to have clear linear structures (like in our case) consistently across different choices of perplexity.
>
> > Everywhere: The "prior" referred to in this paper is not a prior in the Bayesian sense. I suggest a more careful use of terminology.
>
> We thank reviewers for the feedback and agree that a general use of prior can be confusing. It is worth noting that here the "prior" is with respect to the target task, which the model has not seen and learned before modulation or gradient-based optimization. Thus the model is, in fact, a "prior" model, to the target task. We revised the paper to make this more clear.
>
> > Abstract: "augment existing gradient-based meta-learners" You augment a specific variant of gradient-based meta-learning, MAML.
>
> We updated the abstract to make this clear. We believe this modification can be applied to the family of gradient-based meta-learners that seek a parameter initialization and perform gradient steps to adapt to tasks.

---

> ### Author Response · Authors · 2018-11-20
> **Response to AnonReviewer1 (Part 3/3)**
>
>
> > The terminology of "task distribution" and "modes" thereof is used without introduction in the introduction section. The terminology "model-based meta-learning/adaptation" and "gradient-based meta-learning/adaptation" is also used without introduction here. This makes the introduction unnecessarily opaque. Consider the reader who is not familiar with meta-learning papers; they would have a very hard time parsing, for example, the phrase "...this not only requires additional identity information about the modes, which is not always available or is ambiguous when the modes are not clearly disjoint..." (pg. 1).
>
> We revised the introduction and writing to be more clear about the terms we use. The first time the terms “model-based meta-learning” and “gradient-based meta-learning” are used they are introduced with one sentence summaries of their meanings and literary references.
>
> > Further, the terminology "model-based" seems non-standard, and is aliased with the term model-based reinforcement learning (which specifically refers to the set of RL algorithms that make use of a "model" of transition dynamics). Since the paper tackles a reinforcement learning benchmark, this may lead to some confusion.
>
> In the original paper, the term “model-based meta-learning” is explained when it is first mentioned in the introduction and this line of work is presented in the related work section. We believe its meaning is clear in the paper.
>
> > The paper needs to be checked over for English grammar and style.
> > everywhere: "task specific" -> task-specific
> > pg. 3: "relevant but vaguely related skills" this is imprecise
> > pg. 3; "our model does not maintain an internal state" Is the task representation/embedding not an internal state?
> > pg. 3: The episodic training setup, which is standard to meta-learning setups, could be much better described. The MAML algorithm could be given better intuition.
> > Algorithm 1: "infer" is a misuse of terminology that usually refers to an operation in latent variable probabilistic modeling. Since the computation of \tau is purely feedforward, I recommend writing "compute."
> > \tau should be used in some places where v is used instead
>
> We appreciate the reviewer’s advice. We revised the paper to address these points.
>
> [1] Finn et al. “Model-Agnostic Meta-Learning for Fast Adaptation of Deep Networks”, ICML 2017
> [2] Finn et al. ”Probabilistic Model-Agnostic Meta-Learning”, NIPS 2018
> [3] Kim et al. “Bayesian Model-Agnostic Meta-Learning”, NIPS 2018
> [4] Lee and Choi “Gradient-Based Meta-Learning with Learned Layerwise Metric and Subspace”, ICML 2018
> [5] Grant et al. “Recasting Gradient Based Meta-Learning as Hierarchical Bayes”, ICLR 2018
> [6] Nichol et al. “Reptile: a Scalable Meta-learning Algorithm”, arXiv 2018
> [7] Zaheer et al. “Deep Sets”, NIPS 2017

---

### Author Response · Authors · 2018-11-20
**Our General Response**

We thank all reviewers for their constructive feedback. We revised our paper based on reviewers' suggestions and we believe that we addressed most of the concerns. We appreciate the reviewers spending time reading the revised paper in advance. We are more than happy to address further concerns before the end of the rebuttal deadline. Please do not hesitate to let us know for any additional comments on the paper so that we can further improve the paper.

---

### Author Response · Authors · 2018-11-25
**Follow-up Before the Rebuttal Deadline**

We sincerely appreciate the constructive reviews provided by all reviewers. We addressed the concerns in our response and revision. We believe our contributions toward multimodal model-agnostic meta-learning are solid. We would like to kindly ask the reviewers to let us know if there is any further comment towards our revised paper, and we wish to address it before the end of the rebuttal period. Thanks!

---

### Meta-Review · Area_Chair1 · 2018-12-14

**Confidence:** 4
**Recommendation:** Reject

**Metareview:**

This paper proposes a meta-learning algorithm that extends MAML, particularly focusing on multimodal task distributions. The paper is generally well-written, especially with the latest revisions, and the qualitative experiments show some interesting structure recovered. The primary weakness of the paper is that the experiments are largely on relatively simple benchmarks, such as Omniglot and low-dimensional regression problems. Meta-learning papers with convincing results have shown results on MiniImagenet, CIFAR, CelebA, and/or other natural image datasets. Hence, the paper would be more compelling with more difficult experimental settings. In the paper's current form, the reviewers and the AC agree that it does not meet the bar for ICLR.